# Aerosolized In Vivo 3D Localization of Nose-to-Brain Nanocarrier Delivery Using Multimodality Neuroimaging in a Rat Model—Protocol Development

**DOI:** 10.3390/pharmaceutics13030391

**Published:** 2021-03-15

**Authors:** Michael C. Veronesi, Brian D. Graner, Shih-Hsun Cheng, Marta Zamora, Hamideh Zarrinmayeh, Chin-Tu Chen, Sudip K. Das, Michael W. Vannier

**Affiliations:** 1The Department of Radiology and Imaging Sciences, School of Medicine, Indiana University Indianapolis, Indianapolis, IN 46202, USA; braner@iu.edu (B.D.G.); hamzarri@iupui.edu (H.Z.); 2The Department of Radiology, University of Chicago, Chicago, IL 60637, USA; smallgi2002@gmail.com (S.-H.C.); mslewis@uchicago.edu (M.Z.); ccchen3@uchicago.edu (C.-T.C.); mwvannier@gmail.com (M.W.V.); 3The Department of Pharmaceutical Sciences, College of Pharmacy & Health Sciences, Butler University, Indianapolis, IN 46208, USA; skdas@butler.edu

**Keywords:** intranasal delivery, positron emission tomography, hybrid nanoparticles, nose to brain delivery, nanotechnology, fate mapping, molecular imaging, PET/CT, aerosolized drug delivery

## Abstract

The fate of intranasal aerosolized radiolabeled polymeric micellar nanoparticles (LPNPs) was tracked with positron emission tomography/computer tomography (PET/CT) imaging in a rat model to measure nose-to-brain delivery. A quantitative temporal and spatial testing protocol for new radio-nanotheranostic agents was sought in vivo. LPNPs labeled with a zirconium 89 (^89^Zr) PET tracer were administered via intranasal or intravenous delivery, followed by serial PET/CT imaging. After 2 h of continuous imaging, the animals were sacrificed, and the brain substructures (olfactory bulb, forebrain, and brainstem) were isolated. The activity in each brain region was measured for comparison with the corresponding PET/CT region of interest via activity measurements. Serial imaging of the LPNPs (100 nm PLA–PEG–DSPE+^89^Zr) delivered intranasally via nasal tubing demonstrated increased activity in the brain after 1 and 2 h following intranasal drug delivery (INDD) compared to intravenous administration, which correlated with ex vivo gamma counting and autoradiography. Although assessment of delivery from nose to brain is a promising approach, the technology has several limitations that require further development. An experimental protocol for aerosolized intranasal delivery is presented herein, which may provide a platform for better targeting the olfactory epithelium.

## 1. Introduction

Central nervous system (CNS) disorders are often difficult to treat, since many drugs are unable to cross the blood–brain barrier (BBB) [1]. It is projected that ≈100% of large-molecule neurotherapeutics and more than 98% of all small-molecule drugs are effectively inhibited by the BBB [2]. Understanding the mechanism of the delivery and the fate of drugs that can bypass the BBB is an imperative first step in developing viable drug therapies for CNS diseases [3].

Theranostics is a fast-emerging field with the potential to assist with overcoming challenges associated with drug delivery to the CNS. Theranostics merges the potential of the radiological sciences with the delivery of therapeutic agents, thereby promising to advance the goals of personalized medicine [4]. One important aspect of theranostics is that it utilizes multifunctional nanocarriers, which enables simultaneous molecular imaging and targeted therapy delivery by combining a therapeutic drug with an imaging label. Among these nanocarrier agents are biodegradable polymers with acceptable toxicity profiles, biocompatibility, and non-immunogenicity [5]. Polylactide (PLA), polyglycolide (PGA), and polylactide-co-glycolide (PLGA) nanoparticles (NPs) have been the most extensively investigated for drug delivery [6].

Intranasal drug delivery (INDD) is an attractive alternate strategy for circumventing the BBB for local drug delivery to the CNS. It is relatively non-invasive, it avoids first-pass metabolism, and side effects can be minimized because only very small drug concentrations may be needed for treatment. Due to its non-invasive nature, there are reduced risks of infection and disease transmission. Nasal spray formulations are easy to administer and can be performed at home by the patient. Following intranasal delivery, nanocarriers enter the CNS via transcellular and intercellular routes and gain access to the perineural and perivascular fluid channels surrounding the olfactory and trigeminal nerves [7]. Nose-to-brain (N2B) transport may bypass the BBB and provide a conduit for the entry of drugs into the brain [8,9]. The anatomical and cellular structures of the nasal cavity were recently reviewed with tabulation of advantages and limitations for nose to brain delivery [10].

Most CNS drug delivery research involves sacrificing experimental animals and measuring tissue levels using autoradiography, gamma tracer counting, high performance liquid chromatography, and histology, although in vivo imaging of intranasal delivery in both animals and humans is increasing [11]. We chose to focus on positron emission tomography (PET) imaging combined with computer tomography (CT) because of its superior sensitivity and good anatomical detail. Gamma counting was utilized as a reference point for comparison ex vivo.

Aerosolized intranasal drug delivery to target the dorsal nasal cavity in small animals is not a novel concept. However, it is difficult to perform in a reproducible manner in rodents, given their small size and intricate nasal anatomy. A better understanding of intranasal drug delivery in a living animal model will be critical for increasing the translation of drugs to the clinic for nose-to-brain delivery. Herein, we delineate the challenges associated with preclinical small animal aerosolized drug delivery and outline a strategy for improving intranasal delivery using multimodality in vivo imaging.

The purpose of this study was two-fold: (1) To establish the feasibility of preclinical multimodality imaging for intranasal polymeric NP administration via nasal tubing, and (2) to develop a strategy for implementing aerosolized intranasal delivery as an improvement over intranasal delivery via saline tubing.

## 2. Materials and Methods

The methods were carried out in accordance with the relevant guidelines and regulations. All animal studies were approved by the University of Chicago IACUC #72353 (Approval date 27 June 2014) and IU School of Medicine IACUC # (Approval date 19 February 2018).

### 2.1. Nanoparticles

Complete details of the polymer–micellar (polylactide (PLA)–1,2-Distearoylphosphatidylethanolamine (DSPE) nanoparticles prepared and characterized by our team are available in the Appendix A. The PLA–DSPE nanoparticles, coated with polyethylene glycol (PLA–DSPE–PEG), were prepared by Bio Ma-Tek (Bio Materials Analysis Technology Inc., HsinChu City, Taiwan), as previously reported, with modifications [12]. Five milligrams of PLA, 8 mg of DSPE-PEG2000 (1,2-distearoyl-sn-glycero-3-phosphoethanolamine-N-[amino(polyethylene glycol)-2000), and 2 mg of DSPE-PEG2000-NH_2_ were dissolved in 0.5 mL of dichloromethane, and then dropped into 3 mL of double-distilled water. The mixed solution was emulsified over an ice bath for 1 min using a microtip probe sonicator (XL-2000, Misonix, Farmingdale, NY, USA) at a 7 W output. After centrifugation at 13,500× *g* for 10 min, the pellets were discarded, and the nanoparticle suspension was washed three times with a 10% sucrose solution by 30 kD MWCO ultrafiltration (Vivaspin 6, GE Healthcare, Chicago, IL, USA). The surface chemistry was comprised of 80% polyethylene glycol (PEG) chains and 20% polyethylene glycol chains with terminal amino groups (PEG–NH_2_) for further covalent surface modification capability (iron oxide metal, radiotracer tagging, ligand attachment, etc.).

The NPs were characterized in vitro using transmission electron microscopy, dynamic light scattering (DLS), and zeta potential measurements. Full details of this are available in the Appendix A. Images of the nanoparticles were obtained using a transmission electron microscope (TEM; Hitachi model H-7650) with an acceleration potential of 100 kV. The samples were prepared by layering the NP suspension on a copper grid, followed by negative staining for 10 s with a freshly prepared, sterile, filtered 2% (*w*/*v*) uranyl acetate solution. The mean size of the PLA–PEG NPs as determined by TEM was 41.1 nm (*n* = 506). The hydrodynamic diameter and zeta potential of the PLA–PEG NPs were measured using a particle size analyzer (NanoBrook 90Plus, Brookhaven Instruments Corp., Holtsville, NY, USA) and a zeta potential analyzer (NanoBrook ZetaPALS, Brookhaven Instruments Corp., Holtsville, NY, USA) equipped with a 660 nm laser. The measured delay time correlation functions were fitted to a non-negative least squares (NNLS) model to calculate the particle size distribution. The DLS showed a hydrodynamic diameter of 97.1 nm, and the polydispersity (PDI), as measured by the dynamic light scattering technique, was 0.19. Meanwhile, the surface charge (zeta potential) was measured at −36.0 mV.

### 2.2. Nanoparticle Radiolabeling with Zirconium 89

The details of the NP labeling with zirconium 89 are again available in the Appendix A. Briefly, zirconium 89 (^89^Zr) was produced with a cyclotron at Washington University at St. Louis, USA, and overnight shipped to our institution for NP tagging. For the ^89^Zr-labeling, the PLA–PEG NPs were first conjugated with a derivative of desferrioxamine (DFO-Bz-NCS) through amide formation. The PLA–PEG NPs radiolabeled with ^89^Zr (1 mCi) were added to 0.4 mg of PLA–DFO and incubated in pH 7.4 HEPES buffer for 30 min. The radiolabeled PLA–PEG NPs were then purified by centrifugation, and the labeling efficacy was measured with an instant thin-layer chromatography (ITLC) autoradiogram. The radiolabeling activity was 650 µCi per 1 mg of PLA–PEG NPs. We employed the standard operating procedure (SOP Zr-Her 203) from Washington University, USA, for determination of the effective specific activity (ESA) of ^89^Zr by instant thin-layer chromatography (ITLC), which was determined as being a mean ESA of 419 mCi/µmol. ^89^Zr decays by positron emission (23%) and electron capture (77%) to the stable isotope yttrium 89 (^89^Y), and has attractive characteristics for immune-PET applications [13]. Due to a physical decay half-life of 3.3 days, the maximum emission of 897 keV and the average emission of 396.9 keV for its positron emission are likely responsible for the increased spatial resolution of PET images when ^89^Z is used. For further details of the specific activity of ^89^Zr, as well as the radiochemical stability of ^89^Zr-labeled NPs, the reader is again referred to a detailed description in the Appendix A.

### 2.3. Animals

A total of 20 male Sprague–Dawley rats (300–500 g) (Harlan Industries, Indianapolis, IN, USA) were evaluated. The mean age was 12 weeks. The majority of rats were 8 weeks of age (*n* = 8), with the oldest rat at age 25 weeks, which corresponds to a 6-month old rat (18 years old in human years) [14]. The rats were maintained under controlled environmental conditions (23 °C, 12 h light/dark cycle) with free access to standard laboratory chow and tap water prior to the INDD. A total of 20 animals were utilized for this feasibility study, with 6 animals undergoing 2 h of continuous PET/CT imaging (*n* = 3 for INDD experimental group and *n* = 3 for intravenous control group) and 14 animals undergoing gamma counting following brain isolation and tissue dissection (*n* = 3 for 1 h following INDD experimental treatment, *n* = 3 for 1 h following intravenous (IV) control treatment, *n* = 5 for 2 h following INDD experimental treatment, and *n* = 3 for 2 h following IV control treatment).

### 2.4. PET/CT Imaging

The small animal PET/CT imaging studies were carried out on a trimodality preclinical microPET/SPECT/CT imaging system (Trifoil Triumph™, Northridge Tri-Modality Imaging, Inc., Chatsworth, CA, USA). PET images were reconstructed using the ordered-subsets expectation maximization algorithm. CT imaging of the animals was performed, followed by co-registration of the PET images (AMIRA version 3.1 software, FEI Co., Hillsboro, OR, USA). The X-PET detector employs bismuth germanate (BGO) crystals with a full ring geometry. X-PET delivers a spatial resolution of 1.6–2.0 mm full width half maximum (FWHM), a sensitivity above 8%, and an axial field of view (FOV) of ≈12 cm. The PET data were acquired using a 250–750 keV energy window and a 12 ns timing window in the list mode format, which was rebinned into three-dimensional (3D) sinograms corresponding to total and true events, respectively. The animals were imaged for 2 h in the list mode as the data mode immediately following INDD of 100 uCi of ^89^Zr-labeled PLA–PEG NPs.

### 2.5. Nasal Tubing Delivery Imaging Workflow

The rats were anesthetized, followed by INDD or IV injection. For the INDD experiments, each rat was anesthetized with 4% isoflurane and placed in a supine position. Size PE10 flexible tubing was carefully inserted 2 cm deep into the right dorsal nasal cavity and attached to a syringe pump (Model 11, Harvard Biosciences, Inc. (Holliston, MA, USA) calibrated to deliver 30 µL/min. A total of 15 µL of PLA–PEG NPs was delivered over 30 s. A single rat was also administered ^89^Zr alone via the aerosolizer unit. IV controls were established using anesthetized rats maintained with mask inhalation. A tail vein was identified (Figure 1). A ½ cc tuberculin syringe with a 29-gauge needle containing 15 µL of the NP solution (100 µg PLA–PEG NPs with 100 µCi of ^89^Zr tagged to the surface) was injected into the tail vein. The animals were then transferred in the supine position into the PET/CT unit for image acquisition. The acquired images were transferred to a workstation for post-processing (Amira, FEI Co., Hillsboro, OH, USA) with a manual region of interest definition. The regions of interest were constructed in 3D by the investigators to exclude the PET signal from the extracranial structures. PET and CT were co-registered in 3D using the workstation software. This registration process was augmented using a digital version of the Paxinos and Watson rat brain atlas [15] that was scaled and registered to the PET/CT images. The rat brain atlas registration for brain subregion analysis was performed using commercially available software (VivoQuant 2.0, inviCRO LLC, Boston, MA, USA) (Figure 2). Standard uptake values (SUVs) were computed for the brain regions based on the electronic atlas regions of interest.

### 2.6. Ex Vivo Autoradiography

Transcardiac perfusion–fixation was performed on two rat brains, and 1–2 mm sections were made using sagittal precision rat brain slice matrices (Braintree Scientific, Braintree, MA, USA). Tissues were placed on glass slides, air-dried, and positioned against high-efficiency storage phosphor screens for 24 h prior to scanning. Screens were scanned at a 600 dpi resolution using a Bio-Rad FX Pro Plus Molecular Imager (Bio-Rad laboratories, Inc., Hercules, CA, USA).

### 2.7. Ex Vivo Validation of the PET Activity Measurements

After completion of the imaging, the animals were sacrificed, and their brains were removed for independent verification of NP localization. Three brain regions were dissected: The olfactory bulb, brain stem, and forebrain (Figure 1). The forebrain was used to describe the supratentorial cerebral hemispheres without the olfactory bulb. Each sample was placed into a plastic vial for automated activity measurement using a gamma counter (Cobra Quantum 5002 Gamma Counter, Hewlett Packard) (Figure 1). The results were tabulated and compared with the region of interest (ROI) measurements of the SUVs obtained from the corresponding PET scans.

### 2.8. Intranasal Aerosolizer

A functioning prototype of a small animal intranasal drug delivery system (Figure 3A) was developed using a previously published system originally detailed in Piazza et al. [16] with specific modifications made in our laboratory for coupling with a PET system. The aerosolizer was 3D-printed with a central chamber capable of holding up to 50 µL of solution (Figure 3B). Very similar to that described in great detail in Piazza et al., a vertical inlet tunnel communicates with the ceiling of the central chamber where the drug solution is delivered. An anteriorly directed horizontal tunnel leads to the nozzle tip near where the solution is aerosolized. Once the solution is delivered into the central chamber, tubing is fitted over the opening, and over two lateral air chambers that coalesce near the tip of the aerosolizer. An air compressor delivers air to the aerosolizer device through two lateral chambers, and to the input tunnel central reservoir via the attached tubing (Figure 3E). Air forces the drug solution from the central chamber into the horizontal outflow chamber toward the nozzle tip located within the rat’s nasal cavity. At the same time, bi-phasic air flow within the lateral chambers mixes with the solution, causing expansion that introduces instabilities in the liquid–air interface, leading to breakup of the solution and nebulization. The high-pressure air flow aerosolizes the drug solution as soon as it emerges out of the nozzle to form a single controlled spray into the nasal cavity. An anesthetized rat is positioned within the induction chamber, which also houses the aerosolizer affixed to the bite bar (Figure 3E). The central aerosolizer chamber is loaded with 25 µL of NP suspension, which can be delivered hands-free once the anesthetized rat is positioned within the PET unit. The user has computer-based control over the pressures applied via a simple on/off regulator that can control the pulse duration and pressure applied.

### 2.9. Aerosolized Spray Analysis

For optimization of the aerosolizer spray pattern, the droplet size and volumes of drug delivered at differing pressures were evaluated. The testing rig consisted of a platform to which the nasal piece was secured, and a sheet of paper coated in petroleum jelly secured to a stand set at a known distance from the tip of the aerosolizer (Figure 4). The testing was performed for each aerosolizer at 10–16 pounds per square inch (psi) in two psi increments at distances of 5 and 11 mm. The images were photographed with an iPhone 11 and processed in ImageJ. Measurements detailing the location, area, and shape of each drop were exported to MATLAB for analysis. The distances between the aerosolizer and the paper were chosen based on the literature-based measurements [17].

### 2.10. In Vivo PET Imaging of the Aerosolized Intranasal Delivery

An anesthetized rat was positioned within the induction chamber. The central aerosolizer chamber was loaded with 25 µL of NP suspension, which was delivered hands-free once the rat was positioned within the PET unit. Free ^89^Zr in saline solution was administered intranasally to a 300 g rat via the aerosolizer and imaged in real time via PET imaging. The results were of a static compilation of a 30 min acquisition, with most of the tracer remaining within the nasal cavity and a small amount of radiotracer activity being visible within the esophagus.

### 2.11. Statistical Analysis

DICOM image data were post-processed to co-register PET/CT scans and atlas-based image segmentation maps. The ROI measurements and well counts were entered in a spreadsheet (Excel, Microsoft Corp, Redmond, Washington, USA.) These data were evaluated with summary statistics and *t*-tests performed using Stata (StataCorp LP, College Station, TX, USA) and R (https://www.r-project.org/, accessed on 12 August 2015).

## 3. Results

Sagittal, coronal, and horizontal PET/CT images of a representative animal at 1 h after INDD and the image acquisition are shown in Figure 5A. Figure 5B depicts the same animal with the brain removed as a proof of concept to eliminate the influence of scatter from the adjacent nasal cavity. The results were also reproduced using autoradiography with two animals, showing stronger activity in the brain stem, and another showing stronger activity in the olfactory bulb (Figure 6). The data collected for the in vivo imaging represent a composite of 30 min of collections over 2 h (0–30 min, 30 min–1 h, 1–1.5 h and 1.5–2 h after INDD). The ex vivo images and autoradiography are from a single time point, and thus they do not depict exactly the same data for all animals. Post-processing image analysis of the in vivo PET/CT scans yielded the quantitative measures shown in Figure 7 for the datasets obtained after 1 and 2 h delays for comparison between the INDD and IV-treated animals. The amount of activity in the olfactory bulb following INDD was 35.1 or 28.6-fold higher than following IV 1 and 2 h post-treatment, respectively. Additionally, the amount of activity in the brainstem following INDD was 28.9 or 29.6-fold higher than following IV 1 and 2 h post-treatment, respectively. Lastly, the amount of activity in the forebrain following INDD was 11.2 or 7.8-fold higher than following IV 1 and 2 h post-treatment, respectively.

The gamma counter results from the brain subregions are provided from 1 and 2 h following INDD and IV delivery (Figure 8). The subregions correspond with the ROIs defined in the PET/CT scans in both the animals exposed to intranasal NPs and the controls with IV injections of NPs. The amount of activity in the olfactory bulb following INDD was 5.3 or 4.3-fold higher than following IV 1 and 2 h post-treatment, respectively. Additionally, the brainstem activity following INDD was 1.7 or 4.7-fold higher than following IV 1 and 2 h post-treatment, respectively. Lastly, the activity in the forebrain following INDD was 0.8 or 2.2-fold higher than following IV at 1 and 2 h post-treatment, respectively.

### 3.1. Aerosolized Spray Analysis

The spray characteristics were measured based on a time lapse video (Figure 9A), and by collection of aerosolized blue dye onto a filter paper (Figure 9B). The optimal pressure applied for maximizing the percent sprayed from the drug chamber was determined to be 12 PSI (Figure 9D), beyond which higher PSI applications did not result in a greater percentage of drug solution administered. Consistently, 90% of the solution administered to the central chamber was removed from the chamber during aerosolization. The plume characteristics revealed maximum aerosolization centrally within rings 1 and 2 (Figure 9C,E), with a small average size per drop in the central two rings, which had the greatest chance of reaching the olfactory epithelium (Figure 9F). Larger droplets were considered less ideal because of their potential for faster clearance from the nasal cavity. The countable droplets were minimized to 10–12 PSI.

### 3.2. In Vivo PET Imaging of the Aerosolized Intranasal Delivery

An anesthetized rat was positioned within the induction chamber. The central aerosolizer chamber was loaded with 25 µL of the NP suspension, which was delivered hands-free once the rat had been positioned within the PET unit. Free ^89^Zr in saline solution was administered intranasally to a 300 g rat via the aerosolizer and imaged in real time via PET imaging. The results were of a static compilation of a 30 min acquisition, with most of the tracer remaining within the nasal cavity and a small amount of radiotracer activity visible within the esophagus (Figure 10).

## 4. Discussion

Imaging of the INDD-administered nanoparticles (100 nm PLA–PEG–^89^Zr NPs) demonstrated greater activity in the brain after INDD compared to IV delivery (1 and 2 h after treatment). IV administration was used for comparison, since this route labels the blood pool and provides prolonged exposure of the blood–brain barrier. Increased activity in the brain after INDD compared to IV delivery is in keeping with previous reports, especially that of Kozlovskaya et al. in 2014 [18]. We observed much higher levels of activity in the olfactory region and brainstem compared to the forebrain after INDD, suggesting that the NPs entered the brain through the olfactory and trigeminal pathways, as the forebrain is more distal with respect to these two points of entry into the brain. The specific route of administration (whether transneuronal, transcellular, intercellular, or perineural/perivascular) remains unknown. However, the rapid appearance in the brain within 1 h suggests that bulk convection flow through the perineural/perivascular route is a likely contributor.

We cannot say with certainty whether the signal arising from the brain was within the intracellular or extracellular space/cerebrospinal (CSF) spaces, or even partially within the blood pool. However, we did perform intra-cardiac saline perfusion through the left ventricle until the return from the right atrium was clear, indicating removal of the majority of the blood from the intravascular space. During brain isolation, we dissected specific regions of tissue to minimize the contribution from the CSF spaces. Nevertheless, the CSF spaces were too small to discern with the unaided eye. For instance, there is an olfactory cistern coursing through the middle of each olfactory bulb, but the diameter of this CSF space is very small compared to the total volume of brain tissue dissected and analyzed. In the future, we will isolate blood during the intra-cardiac perfusion and assess it with a gamma counter to determine whether there is a contribution of activity from the brain’s intracranial circulation. We will also employ a method of collecting CSF from rats via access to the cisterna magna using the technique developed by Nirogi et al. in 2009 [19]. This is a technically demanding, but important question to answer in future experiments.

Independent verification of the PET/CT imaging findings was provided by gamma counter readings, with which the activity in the brain from the ^89^Zr radiolabeling was greater in the intranasally exposed compared to IV-injected animals. This observation suggests that advanced imaging may be useful for understanding how best to administer, track, develop, and understand the in vivo behavior of NP agents in the CNS. A clear increase in brain activity occurred, as assessed with the gamma counter from 1 h to 2 h. Although the activity in the brain following INDD was increased in the PET results compared to following IV administration, the difference in activity between the 1 and 2 h time points by PET could not be ascertained following INDD within the current sample size. This may be from the lower sensitivity of PET relative to gamma counting. Unlike in the gamma studies (where we removed the brain from the skull and separated the brain from the nasal cavity and oropharynx), we were not able to completely eliminate the possibility that the higher signal activity seen in the olfactory bulb and brainstem in the in vivo studies was at least partially because of the high activity coming from the adjacent nasal cavity (for the olfactory bulb) and oropharynx (for the brainstem). This may also explain the lower values in the olfactory bulb relative to the brain stem on PET, yet higher values in the olfactory bulb compared to brain stem gamma counting results. The mean beta annihilation range in water for ^89^Zr can occur up to 1.34 mm from the actual location of the tracer [20]; thus, not all of the signal in the brain next to the nasal cavity and oropharynx, namely, the olfactory region, can be assured as having arisen from within the brain parenchyma. That limits full quantitation in these regions.

The issue of signal spillover (scatter artifact) from the nasal cavity is a very important issue. There is an inherent limitation to the spatial resolution in PET imaging that will make accurate quantitation of signal at the cribriform plate difficult. On the contrary, it has been known for some time that substances flow between the nasal cavity and the brain peri-/para-vascular spaces, especially at the base of the brain. This might perhaps be viewed more as a continuum through the porous cribriform plate rather than an actual bony barrier physically separating these two compartments. The brain’s ventricular system and subarachnoid spaces are also in communication with the peri-/para-vascular spaces (and hence the nasal cavity). Items injected into the ventricles will subsequently be detected in the nasal cavity and from there travel as distally as the neck [21]. In a study by Bedussi et al. in 2015, dyes injected into the brain parenchyma (striatum) and cisterna magna were also found to be in continuum with the extracellular spaces, basilar cisterns, and nasal cavity [22]. This may also explain the observation that although the NPs were instilled in a single nasal cavity, the signal was visible roughly equally in both olfactory bulb regions, although the possibility that this can be explained by spillover cannot be excluded. Future studies will assess for the presence of nanoparticles in the olfactory tissue using transmission electron microscopy to validate entry of the NPs into the brain and to exclude the possibility that the signal merely represents the radiotracer signal. Regardless, to the best of our knowledge, there are no preclinical studies that have utilized highly sensitive PET/CT dual modality imaging to assess nose-to-brain delivery in a living animal.

Interestingly, we observed greater activity in the brain stem relative to the olfactory region using PET at both time points (Figure 7). This pattern was reversed in the gamma counter studies, where the signal in the olfactory bulbs was higher than that in the brain stem at 1 and 2 h post-treatment (Figure 8). We believe this may reflect the importance of distributing the nanoparticles uniformly in the dorsal superior nasal cavity, which we propose would be an added benefit of aerosolized intranasal delivery over liquid solution delivery through nasal tubing. This finding also points to the importance of understanding and optimizing the adhesion of drugs or NPs within the nasal cavity, which in turn increases retention and absorption [23]. This variability in uptake of the olfactory bulb relative to the brain stem was also demonstrated in the autoradiography study (Figure 6), where one rat had greater uptake in the nasal cavity, and the other had stronger uptake in the brainstem.

Numerous publications have demonstrated quantitative intranasal delivery to the olfactory region, brainstem, and deeper brain regions in an ex vivo manner without the use of dynamic imaging (see reviews by Lochhead et al. in 2012 [7] and Kozlovskaya et al. in 2014 [18]). A few studies have shown initial imaging in living animals, mostly using gamma scintigraphy [24]. However, direct visualization of the activity in the brain in all of these studies was limited by the lack of three-dimensional cross-sectional anatomic correlates. While we are able to demonstrate a differential uptake of activity in the brain after INDD compared to IV administration, we did not perform an area under the curve assessment to determine the direct targeting efficiency, which is becoming the standard of quantitative nose-to-brain drug delivery. This will be adapted in future work with validation studies over longer periods of time.

Since our goal was to establish the feasibility of multimodality INDD imaging of NPs, we chose to image the rats at 1 and 2 h post-administration. The data acquisition times used for this set of experiments were prolonged and not ideal for kinetics faster than the first 30 min following INDD. An additional rationale for the imaging sampling times was based on the work of Born et al. in 2002, a frequently cited human clinical study highlighting a nose-to-brain pathway [25]. Three clinically relevant peptides, namely, melanocortin(4–10) (MSH/ACTH(4–10)), vasopressin, and insulin, were delivered intranasally, followed by CSF detection via an intrathecal catheter. These were compared against an intranasally delivered placebo treatment. Although the detected levels of the three peptides were increased compared to baseline in the CSF as early as 10 min after INDD, it took between 30 and 80 min to reach significantly increased concentrations above baseline. The concentrations of MSH/ACTH(4–10) and vasopressin were still above those in the placebo-treated subjects at 100–120 min after administration. More recently, intranasal oxytocin administration was not detected in significant enough amounts compared to IV delivery until 75 min after INDD [26]. While CSF detection does not imply delivery to the brain parenchyma, it is an indication that it takes at least this long for the drug compounds to reach the brain tissue via the perineuronal/perivascular/subarachnoid spaces. In other animal studies, estimates of peak brain delivery following INDD (including nanoparticles) range between 20 and 60 min to reach the olfactory bulb and between 1 and 4 h to reach the other brain regions (including the brainstem) [7].

It is ubiquitous in the literature that a percentage of nasally delivered nanoparticles in a liquid solution will inevitably be swallowed, since ciliary motion sweeps substances out of the nasal cavity and toward the oropharynx. This was also found to be the case in our initial aerosolized delivery assessment with ^89^Zr. We performed whole body ex vivo biodistribution studies using a human clinical PET/CT scanner following both INDD and IV delivery of the same 100 nm PLA–DSPE–PEG NPs at 30 min and 1 h [27]. As expected, the gastrointestinal tract accounted for a major fraction of the nanoparticles’ localization after being swallowed, and only a small fraction of the nanoparticles could be seen in the brain. Subregional brain delivery was not assessed, since we needed to use a clinical PET/CT scanner to assess the full body biodistribution. A more clinically relevant way to deliver drugs/NPs intranasally would be to use an intranasal aerosolizer rather than a liquid solution in tubing administered through a syringe. According to Piazza et al. [16], the use of aerosolized NPs could increase the amount of NPs that bind to the nasal epithelium via improved nasal cavity mucosal surface coverage, increased state of dispersion, and enhanced impact force of NP–epithelium collisions. This would aid in the capture and subsequent transport of NPs by and then through the mucosal layer. The end result of the greater transnasal absorption would be a reduction in the loss of NPs into the respiratory or gastrointestinal tracts.

This study has several limitations that need to be considered. First, the sample size was small, so future experiments with variations in the type of NPs, doses, and administration methods (such as aerosol spray) will expand our knowledge base. Second, complex multifunctional NPs are subject to metabolic processes that may alter their compositions and separate their component parts. Third, our experiments were limited to PET/CT with gamma counter validation. Additional modalities are available that can be used to explain various aspects of nanoparticle fate, including MRI and optical fluorescence. Fourth, the time course of our study was limited to 2 h, so this limited time scale cannot fully assess NP pharmacokinetics in the CNS. Fifth, there was no histological confirmation or autoradiographic evidence to show the fate of the NPs at a cellular level. The characterization of NPs, especially if intended for ultimate human use, is a complex, technically demanding, and rapidly maturing specialty area. Our NPs received a limited laboratory characterization that must be augmented with comprehensive distribution, metabolism, and toxicological data to more fully understand the biological behavior of the agent. Finally, we did not attempt to show any direct drug effect associated with INDD, but instead limited our attention to the short-term distribution of NPs and to a feasibility demonstration of imaging in the administration phase. Our future work is intended to more thoroughly evaluate the NPs and their mechanism of transport, their time course, and their fate using PET/CT; to establish improvements in image post-processing; and to test other imaging modalities with detailed independent validation of imaging findings.

In conclusion, we demonstrated that radiolabeled NPs administered intranasally via liquid form in nasal tubing can enter the brain with a sufficient concentration to be detected by PET/CT imaging within 1 h, validated using tissue dissection and a gamma counter. These findings indicate that the fate of radiolabeled NPs administered via the intranasal route can be determined with PET/CT imaging in experimental animals (rats). We proposed an improvement on nasal tubing intranasal delivery via a recently developed promising rat aerosolized delivery protocol, but further studies are needed to demonstrate superiority of aerosolized delivery over nasal tubing delivery.

## Figures and Tables

**Figure 1 pharmaceutics-13-00391-f001:**
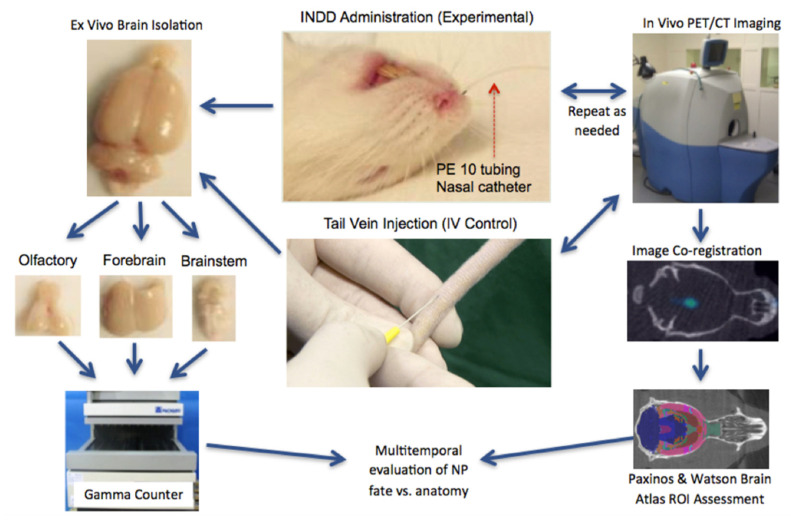
Intranasal drug delivery (INDD) workflow followed by positron emission tomography/computer tomography (PET/CT) imaging at 1 and 2 h. Each rat was placed into a customized induction chamber and anesthetized; and nanoparticles (NPs) were delivered intranasally for the experimental portion of the study or by IV via the tail vein for the control portion of the study. The rat was transferred into the PET/CT unit for imaging at various time points. The acquired images were used to manually define the three-dimensional (3D) regions of interest (ROIs). The intracranial intensity data from the PET and the corresponding CT scan were co-registered and imported into post-processing image analysis software for brain segmentation and subregion measurement. After imaging, the brain of the animal was isolated, and the brain subregions were dissected into the olfactory bulb, brainstem, and forebrain. The tissue was placed in a gamma counter for quantitative subregional assessment. Finally, the in vivo and ex vivo data were analyzed and compared for multi-temporal evaluation of the nanoparticles’ fate within the various anatomical regions of the brain.

**Figure 2 pharmaceutics-13-00391-f002:**
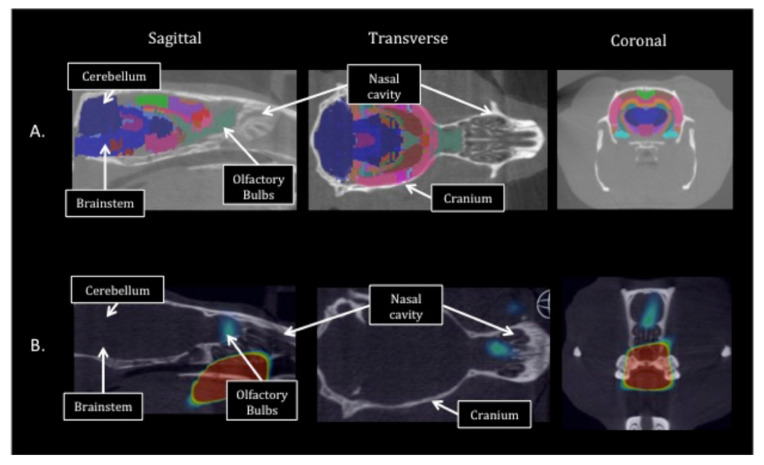
PET/CT image segmentation and labeling for brain subregion definition. The 3D regions of interest were defined for the brain subregions based on the Paxinos and Watson rat brain atlas. A total of 13–64 brain regions and subregions were defined using this automated software. Sagittal, horizontal, and coronal CT images (**A**) through the rat brain with the corresponding PET/CT images (**B**) for the same animal are shown. Example brain regions that were analyzed include the olfactory region, hypothalamus, septal region, amygdala, hippocampus, cortex, thalamus, brainstem, ventricles, cerebellum, and rostral spinal cord.

**Figure 3 pharmaceutics-13-00391-f003:**
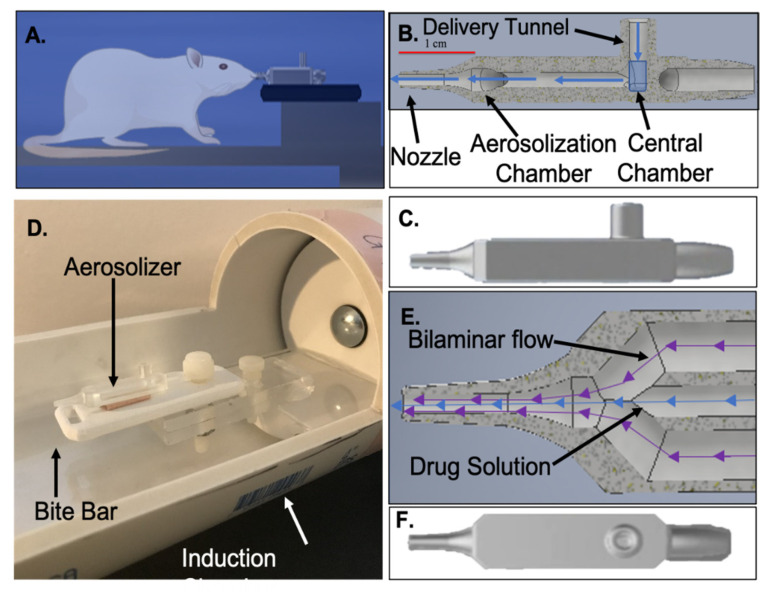
Intranasal aerosolizer set up. (**A**) Schematic of the aerosolizer relative to a cartoon of a rat. (**B**) Sagittal view of the internal component of the aerosolizer with a central chamber, delivery tunnel, aerosolization chamber, and nozzle in a manner very similar to Piazza et al., [16]. (**C**) Sagittal outer view of the aerosolizer. (**D**) Aerosolizer attached to the induction chamber tooth bar apparatus. (**E**) Top-down view of the internal component of the aerosolizer at the nozzle. The direction of the outer bilaminar flow of air (purple arrows) is shown, which then mixes with the central drug solution (blue arrows) to aerosolize the solution. (**F**) Top-down view of the outer aerosolizer.

**Figure 4 pharmaceutics-13-00391-f004:**
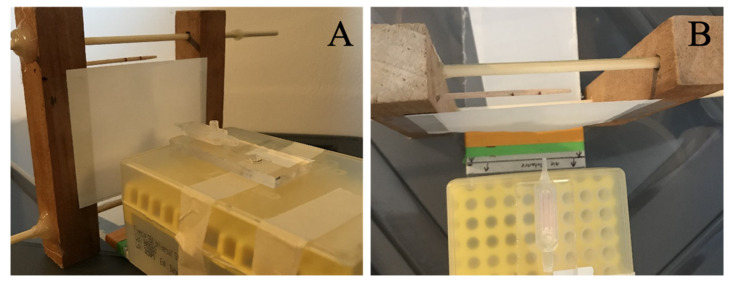
Aerosolized spray analysis. (**A**) Side-view and (**B**) top-view of the aerosolizer testing apparatus. The first edge of green tape represents 5 mm from the aerosolizer tip, and the first edge of orange tape represents 11 mm from the aerosolizer tip. The white paper was coated in petroleum jelly so that droplets could be seen. The toothpick at the top of the paper was marked at a known 1 cm for use later in image analysis.

**Figure 5 pharmaceutics-13-00391-f005:**
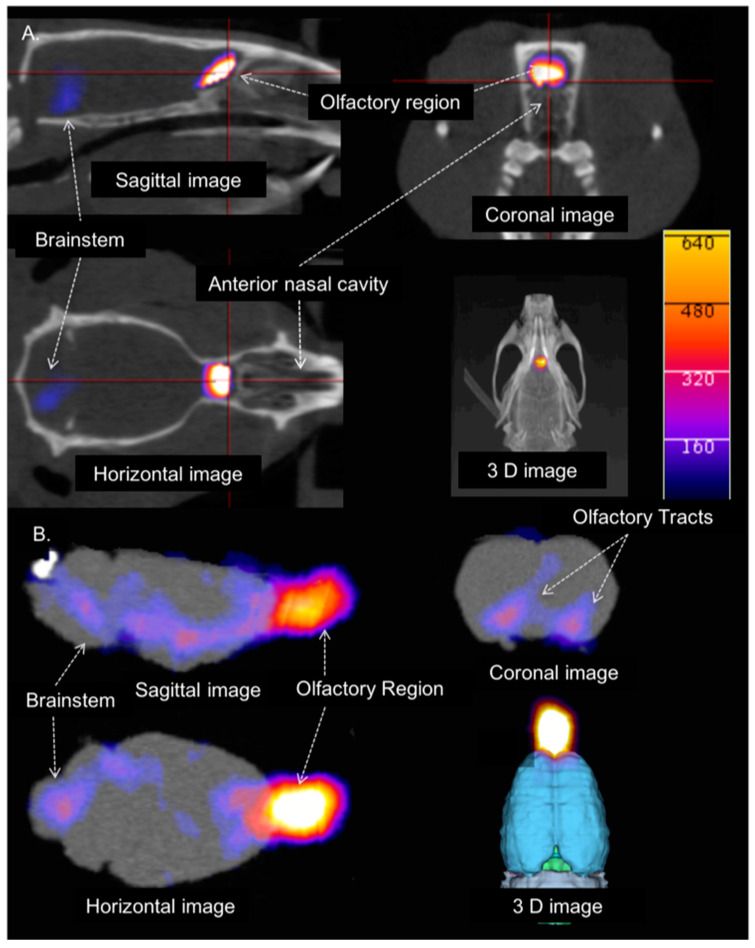
PET/CT brain activity at 1 h following INDD. (**A**) Sagittal, coronal, axial, and 3D images demonstrating the PET activity in the olfactory and brainstem regions. Oropharyngeal activity outside the intracranial ROI was suppressed. (**B**) Sagittal, coronal, axial, and 3D images demonstrating the PET activity in the olfactory and brainstem regions. Increased activity was found in the olfactory bulb, and to a lesser extent, in the basal forebrain structures, brainstem, and trigeminal ganglion. The scale bar intensity ranges from 0 to 640. By changing the upper and lower limits of the color scale, we can observe lower uptake distributions in the brain more readily.

**Figure 6 pharmaceutics-13-00391-f006:**
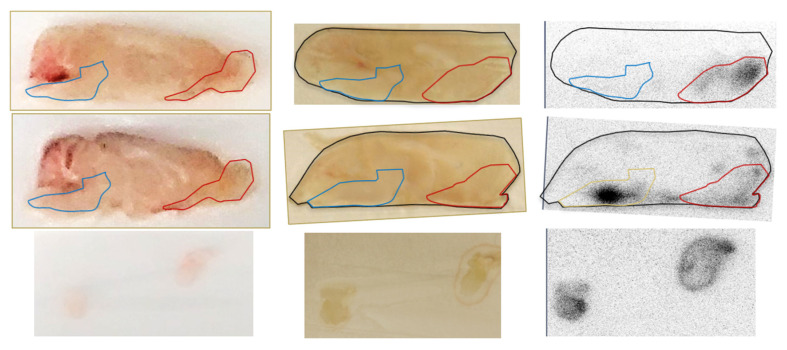
Ex vivo autoradiography of right and left paramedian sagittal brain slices at 2 h following INDD of polylactide (PLA) NPs. Two rat brains were made into 1–2 mm sections using rat brain slice matrices. The far left and middle images of the top and middle rows represent sagittal sections at 2 h following intranasal delivery of zirconium 89 (^89^Zr)-labeled NPs before and after positioning against phosphor screens for 24 h prior to scanning. The far-right images are the radioactivity read outs collected on the film. The bottom row represents the trigeminal nerves isolated from a single rat with radioactivity present in this tissue as well.

**Figure 7 pharmaceutics-13-00391-f007:**
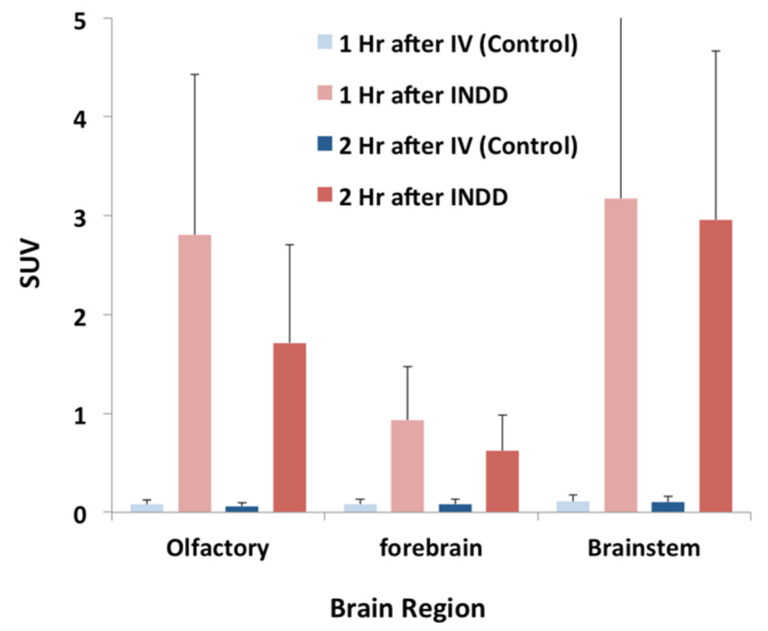
Brain region activity at 1 and 2 h following INDD of 100 nm PLA–polyethylene glycol (PEG)–^89^Zr NPs as detected by PET/CT in vivo. The standard uptake values (SUVs) were calculated using the Vivoquant ROI Paxinos and Watson Brain atlas plug-in to determine the differential activity in various regions throughout the brain. In particular, the signal activity was highest in the olfactory region and brainstem following INDD of the PLA–PEG–^89^Zr NPs. Relatively less activity was seen in the forebrain.

**Figure 8 pharmaceutics-13-00391-f008:**
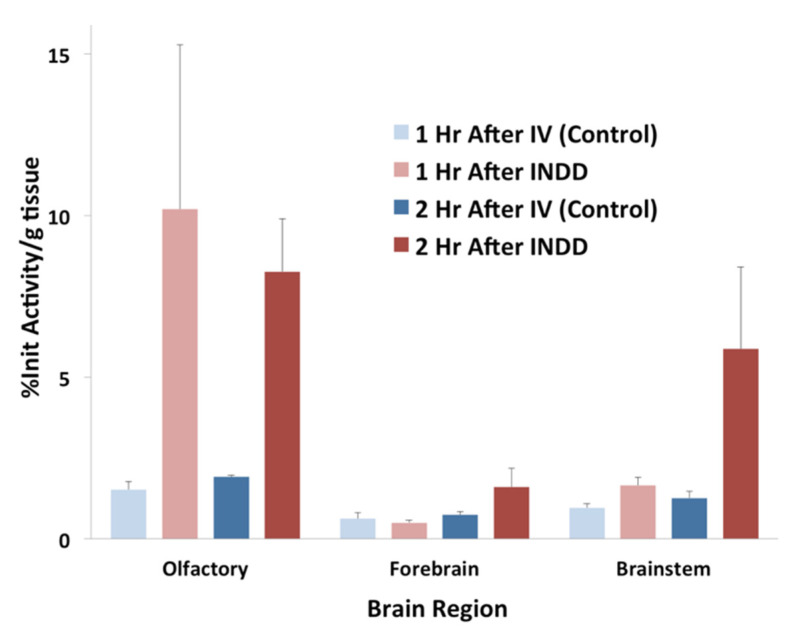
Quantitative brain region activity at 1 and 2 h following INDD of 100 nm PLA–PEG–^89^Zr NPs, as detected by gamma counting ex vivo. The percentage of initial activity per gram of tissue was calculated using a standard gamma counter to determine the differential activity in various regions throughout the brain. Similar to the PET/CT data, the signal activity was highest in the olfactory region and brainstem following INDD of the 100 nm PLA–PEG–^89^Zr NPs. Relatively less activity was seen in the forebrain.

**Figure 9 pharmaceutics-13-00391-f009:**
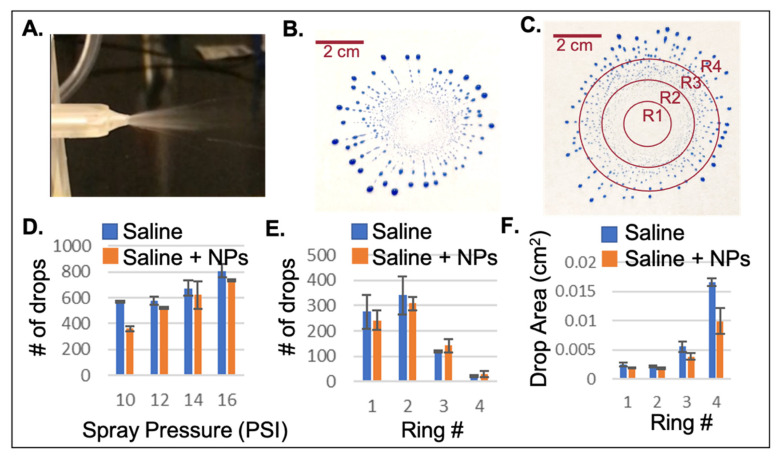
Optimization of the spray characteristics for aerosolized drug delivery to physiologically small animals. (**A**) Photo of the appearance of the aerosolized solution using a time lapse video. (**B**) Example plume captured using blue dye on a filter paper backdrop. (**C**) Separation of the plume into 4 successively larger rings (R1, R2, R3, and R4), analyzed using MATLAB. (**D**) Analysis of the spray pressure and number of drops generated. Ideal conditions would minimize the number of drops. A total of 10–12 PSI was the optimal pressure to emit a minimum of 90% of the solution from the chamber in physiological conditions (data not shown). (**E**) The plume characteristics revealed maximum aerosolization centrally within rings 1 and 2. (**F**) The aerodynamic diameters of the droplets within R1, R2, and R3 were approximately 5, 50, and 500 μm, respectively. The majority of the R1 droplets were expected to reach the olfactory region.

**Figure 10 pharmaceutics-13-00391-f010:**
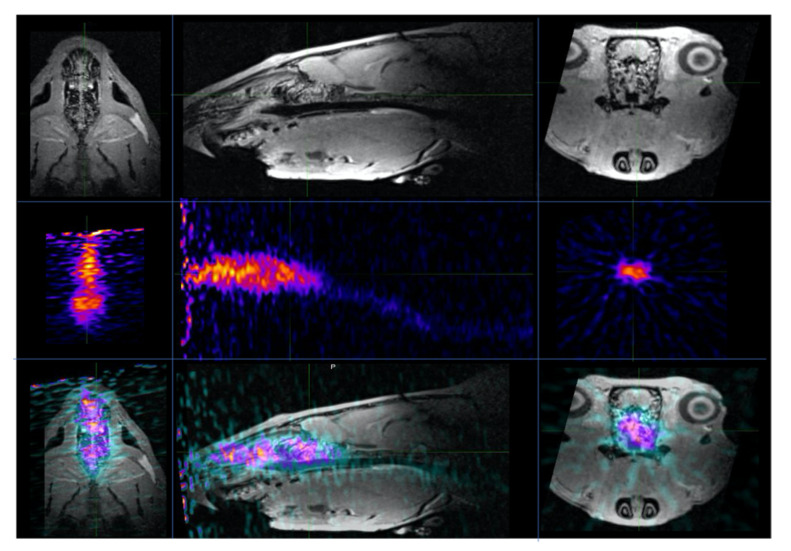
In vivo PET imaging of the aerosolized intranasal delivery. Horizontal, sagittal, and coronal images of a rat following an anatomic magnetic resonance imaging (MRI), functional ^89^Zr PET, and MR/PET fusion. Radiotracer activity was distributed throughout the nasal cavity, with trace activity within the esophagus. Activity was not visualized above the background in the brain or lungs.

## Data Availability

The data in the form of excel sheets presented in this study are available on request from the corresponding author.

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
