# Peer review of "Aerosolized In Vivo 3D Localization of Nose-to-Brain Nanocarrier Delivery Using Multimodality Neuroimaging in a Rat Model—Protocol Development"

_pharmaceutics, 2021, doi:10.3390/pharmaceutics13030391_

Round 1

Reviewer 1 Report

I thank the Authors for presenting an interesting and novel study on intranasal drug delivery and theranostics in an animal model. I did not find any specific issues in the manuscript, as the paper clearly presents a well-designed study.

Author Response

We thank Reviewer 1 in return for a very supportive analysis and comment. 

Reviewer 2 Report

In this manuscript the authors demonstrated that radiolabeled nanoparticles administered in- tranasally via liquid form in nasal tubing can enter the brain with a sufficient concentration to be detected by PET/CT imaging within 1 h, and validated using tissue dissection by a gamma counter. These findings indicate that the fate of radiolabeled nanoparticles administered via the intranasal route can be determined with PET/CT imaging in rats. In the second part of the study aerosolized NP delivery protocol was shown, but further studies are needed to demonstrate superiority of aerosolized delivery over nasal tubing delivery.

The multimodality INDD imaging study is impressive and well documented. The limitations are listed by the authors themselves.

Just some small comments and requests raised:

The animals used in this study were very different in age (bodyweights 300-500 g). The BBB permeability and the degree of brain penetration of compounds correlates with the age of the animals. Please give the exact age of the animals.

Although the nose-to-brain delivery route bypass the BBB for the most part, but there is a certain amount of NP-s that penetrates the mucosal capillaries and gastrointestinal and respiratory tracts. A whole body picture should be added about the peripheral distribution of the tracer.

A further study is suggested for a next manuscript to test the transporter interactions of the used composition at the nasal cavity, and also a combination therapy can be tested by efflux transporter inhibitors (not for this article!!).

A fundamental literature should be added to the introduction:

https://pubmed.ncbi.nlm.nih.gov/30449731/

Author Response

Please see the attachment since we have images we would like to show as well.

Reviewer 3 Report

This paper describes an aerosolized nasal delivery strategy for rodents using multimodality in vivo imaging. The development of this method is important for elucidating intranasal drug delivery in living animals. However, this method has some limitations and there are still some parts that need to be examined in the future. The weakness of this study is that the route of transport of the drug from the nasal to the brain has not been clarified. As the authors argued, it is difficult to identify pathways such as transneurons, transcells, intercellular, perineural / perivascular, etc. Furthermore, the involvement of the cerebrospinal fluid route does not appear to be clear. Nevertheless, this paper is novel and original, and would be suitable for this journal if the following points were clarified.

Major Point

Imaging of LPNP delivered intranasally showed increased activity in the brain 1 and 2 hours after intranasal drug delivery (INDD) compared to intravenous administration. In Figure 7, the activity of the brain stem obtained by PET/CT imaging was higher than that of the olfactory bulb region. However, in Figure 8, the olfactory bulb region obtained by the gamma counting test was higher than the brain stem at 1 h and 2 h after the treatment.

The authors need to discuss this contradictory result in terms of the importance of uniform particle distribution from the viewpoint of the device and the relationship between particle adhesion and retention from the viewpoint of the nasal structure.

Author Response

We would like to thank all three reviewers for their time and expertise. Their reviews have led to a notable improvement in the manuscript that we hope will now be acceptable for publication. Please see our individual responses below. 

Response to Reviewer 3 Comment

Comment:

In Figure 7, the activity of the brain stem obtained by PET/CT imaging was higher than that of the olfactory bulb region. However, in Figure 8, the olfactory bulb region obtained by the gamma counting test was higher than the brain stem at 1 h and 2 h after the treatment. The authors need to discuss this contradictory result in terms of the importance of uniform particle distribution from the viewpoint of the device and the relationship between particle adhesion and retention from the viewpoint of the nasal structure.

Response:

This is a very important point raised by the reviewer. We had alluded to this concept in the discussion on page 14 but have expanded with a high-quality reference to elaborate on this as requested. We hope this brings more clarity.

We acknowledge that the sample sizes were very small, mostly n=3. A large amount of unexplained variability is present that may be due to measurement error, difficulty in dissection, or other. We don’t attempt to draw final conclusions but claim that the approach appears feasible and further work is needed to control variability and achieve statistically significant results.

Here is the paragraph that we revised and added to in the discussion section of the manuscript (page 14, lines 448-459:

Interestingly, we observed greater activity in the brain stem relative to the olfactory region using PET at both time points (Figure 7). This pattern was reversed in the gamma counter studies, where the signal in the olfactory bulbs was higher than that in the brain stem at 1 and 2 h post-treatment (Figure 8). We believe this may reflect the importance of distributing the nanoparticles uniformly in the dorsal superior nasal cavity, which we propose would be an added benefit of aerosolized intranasal delivery over liquid solution delivery through nasal tubing. This finding also points to the importance of understanding and optimizing the adhesion of drugs or NPs within the nasal cavity, which in turn increases retention and absorption [22]. This variability in uptake of the olfactory bulb relative to the brain stem was also demonstrated in the autoradiography study (Figure 6), where one rat had greater uptake in the nasal cavity and the other had stronger uptake in the brainstem.

Reference [22] was added to this paragraph as a nice review performed by Sonvico et al. that we hope will help drive this point home to the reader.

Sonvico, F.; Clementino, A.; Buttini, F.; Colombo, G.; Pescina, S.; Staniscuaski Guterres, S.; Raffin Pohlmann, A.; Nicoli, S. Surface-Modified Nanocarriers for Nose-to-Brain Delivery: From Bioadhesion to Targeting. Pharmaceutics 2018, 10, doi:10.3390/pharmaceutics10010034.  

Round 2

Reviewer 3 Report

Authors have revised the manuscript according to the comments made previously sufficiently.